# Immunological Properties of Atopic Dermatitis-Associated Alopecia Areata

**DOI:** 10.3390/ijms22052618

**Published:** 2021-03-05

**Authors:** Reiko Kageyama, Taisuke Ito, Shiho Hanai, Naomi Morishita, Shinsuke Nakazawa, Toshiharu Fujiyama, Tetsuya Honda, Yoshiki Tokura

**Affiliations:** 1Department of Dermatology, Hamamatsu University School of Medicine, Hamamatsu 431-3192, Japan; reikok@hama-med.ac.jp (R.K.); 41237726@hama-med.ac.jp (N.M.); szawa@hama-med.ac.jp (S.N.); fujiyama@hama-med.ac.jp (T.F.); hontetsu@hama-med.ac.jp (T.H.); 2Department of Dermatology, Seirei Hamamatsu General Hospital, Hamamatsu 431-3192, Japan; hanaishiho@gmail.com; 3Department of Cellular & Molecular Anatomy, Hamamatsu University School of Medicine, Hamamatsu 431-3192, Japan; tokura@hama-med.ac.jp

**Keywords:** alopecia areata, extrinsic atopic dermatitis, intrinsic atopic dermatitis, IL-13, IFN-γ

## Abstract

Alopecia areata (AA) is regarded as a tissue-specific and cell-mediated autoimmune disorder. Regarding the cytokine balance, AA has been considered a type 1 inflammatory disease. On the other hand, AA often complicates atopic dermatitis (AD) and AD is regarded as type 2 inflammatory disease. However, the immunological aspects of AA in relation to AD are still poorly understood. Therefore, we aim to clarify the immunological properties of AD-associated AA. In this study, we performed comparative analysis of the expression of intracytoplasmic cytokines (IFN-γ, IL-4, and IL-13), chemokine receptors (CXCR3 and CCR4) in peripheral blood which were taken from healthy controls, non-atopic AA patients, AA patients with extrinsic AD, and AA patients with intrinsic AD by flowcytometric analysis. We also compared the scalp skin samples taken from AA patients with extrinsic AD before and after treatment with dupilumab. In non-atopic AA patients, the ratios of CD4+IFN-γ+ cells to CD4^+^IL-4^+^ cells and CD4^+^IFN-γ^+^ cells to CD4^+^IL-13^+^ cells were higher than those in AA patients with extrinsic AD. Meanwhile, the ratio of CD8^+^IFN-γ^+^ cells to CD8+IL-13+ cells was significantly higher in the non-atopic AA than in the healthy controls. In AA patients with extrinsic AD, the skin AA lesion showed dense infiltration of not only CXCR3+ cells but also CCR4^+^ cells around hair bulb before dupilumab treatment. However, after the treatment, the number of CXCR3^+^ cells had no remarkable change while the number of CCR4^+^ cells significantly decreased. These results indicate that the immunological condition of AA may be different between atopic and non-atopic patients and between extrinsic and intrinsic AD patients. Our study provides an important notion that type 2 immunity may participate in the development of AA in extrinsic AD patients. It may be considered that the immunological state of non-atopic AA is different from that of atopic AA.

## 1. Introduction

Alopecia areata (AA) is regarded as a tissue-specific and cell-mediated autoimmune disorder. Melanin-generating anagen hair bulbs maintain an immuno-tolerated milieu, i.e., hair follicle immune privilege (HF-IP) [1,2,3]. AA autoantigens, such as trichohyaline, tyrosinase, and tyrosinase-related protein 1/2 (TRP1/2), are immunologically protected from autoimmune attacks during the anagen phase [4]. There are several factors that contribute to this protection, as represented by the lack of major histocompatibility complex (MHC) class I in the proximal outer root sheath (ORS) and matrix cells.

Regarding the cytokine balance, AA has been considered a type 1 inflammatory disease. In the AA model mouse C3H/HeJ, interferon-γ (IFN-γ) is regarded as a key cytokine in the pathogenesis of AA [5,6,7]. Eight-week-old female C3H/HeJ mice injected with a high dose of murine IFN-γ showed AA-like hair loss after 36 days of injection [5]. Furthermore, AA was not induced in IFN-γ-/- mice by the grafting of lesional AA skin from AA-affected mice [8]. In human, IFN-γ-producing cells were detected more frequently in the perifollicular infiltrate of AA lesions than that of healthy skin [9], and the levels of serum Th1 cytokines were increased in AA patients compared to healthy controls [10].

Atopic diathesis, including allergic rhinitis, asthma, and/or eczema, is prevalent at a frequency as high as 38.2% in AA patients, followed by contact dermatitis, mental health problems, and autoimmune diseases [11,12,13,14,15,16]. Seven cross-sectional studies have reported that the prevalence of both AA and atopic history in adults ranges from 22% to 38% when confounding variables are considered [11,12,13,14,15,16]. AD is primarily a Th2-driven disease with increased levels of interleukin (IL)-4, IL-5, IL-13, and IL-31 [17,18,19]. Although AD and AA are commonly encountered skin disorders, the immunological aspects of AA in relation to AD are still poorly understood.

AD displays different clinical presentations and has been characterized by several types depending on age, ethnicity, immunological aspects and the underlying pathomechanisms. For example, Nettis et al. categorized AD into 3 different types such as persistent form in which AD appears in childhood and is maintained (with its chronic-recurrent course) until adulthood, by relapsing form with childhood onset of the disease and a relapse of symptoms after some symptom-free years; and adult-onset AD in which the disease first appears in adulthood [20]. AD also characterized by clinical phenotypes such as lichenified/exudative flexural dermatitis alone and associated with portrait dermatitis, nummular eczema-like phenotype and prurigo nodularis-like pattern [20]. In addition, AD can be categorized into the extrinsic and intrinsic types by levels of IgE and skin barrier function [21]. Extrinsic AD exhibits high total serum IgE levels with skin barrier dysfunction, whereas intrinsic AD shows normal total IgE values with relatively normal skin barrier function. Understandings of the pathogenesis of AD, the innovative treatments, such as dupilumab, tofacitinib, baricitinib, crisaborole, nemolizumab, apremilast and Lebrikizumab, have been available for severe AD [22]. In these novel treatments for AD, dupilumab is a fully human antibody that recognizes IL-4Rα and blocks both the IL-4 and IL-13 receptor signaling, whose downstream is the JAK-STAT pathway [23,24]. Recent observations have shown that dupilumab exerts therapeutic effectiveness not only for AD, but also for AA [11,16,25,26,27,28,29,30], raising a possibility that the inflammatory aspects of these two diseases are enigmatically related to each other. In this study, we examined the immunological conditions in patients with AA associated with extrinsic or intrinsic AD in a comparison with non-atopic AA patients. We then sought to investigate the therapeutic effect of dupilumab on AA associated with AD. AA was successfully treated with dupilumab in patients with extrinsic AD, while no remarkable effect on AA was seen in one intrinsic AD patient. Immunological alterations between pre- and post-therapy were also monitored in these patients. Our study provides some insights for the pathogenesis of AA in relation to AD.

## 2. Results

### 2.1. Type 2 Polarization in Circulating T Cells from AA Patients with Extrinsic AD

AA has been regarded virtually as a type 1 inflammatory disease with dominant expression of IFN-γ [10]. We first examined the frequencies of circulating Th1/Tc1 and Th2/Tc2 cells in patients with non-atopic AA, extrinsic or intrinsic AD-associated AA, and control normal subjects by measuring the percentages of CD4^+^IFN- γ^+^ cells (Th1), CD4^+^IL-4^+^ cells (Th2), CD4^+^IL-13^+^ cells (Th2), CD8^+^IFN-γ^+^ cells (Tc1), CD8^+^IL-4^+^ cells (Tc2), and CD8^+^IL-13^+^ cells (Tc2). The cytokine expression was evaluated by intracellular cytokine staining and subsequent flow cytometry. Due to high variations in these percentages, we expressed the type 1 and 2 balance as the ratio of Th1(Tc1) to Th2 (Tc2). In non-atopic AA patients, the ratios of CD4^+^IFN-γ^+^ cells to CD4^+^IL-4+^+^ cells (Figure 1a) and CD4^+^IFN-γ^+^ cells to CD4^+^IL-13^+^ cells (Figure 1b) were significantly higher than those in AA patients with extrinsic AD (*p* = 0.0048). In the ratio of CD8^+^IFN-γ^+^ cells to CD8^+^IL-4^+^ cells, there were no significant differences between any groups (Figure 1c). However, the ratio of CD8^+^IFN-γ^+^ cells to CD8^+^IL-13^+^ cells was significantly higher in the non-atopic AA than in the healthy controls (*p* = 0.0011) (Figure 1d). The ratio of CD8^+^IFN-γ^+^ cells to CD8^+^IL-13^+^ cells was also significantly higher in the intrinsic atopic AA than in the extrinsic AD (*p* = 0.0321) (Figure 1d). In these non-atopic AA patients, the frequency of IFN-γ-producing Tc1 cells was inversely correlated with the serum IgE levels (Figure 1e). Chemokine receptor CXCR3 and CCR4 were used as alternative markers for Th1/Tc1 and Th2/Tc2 cells, respectively. The CXCR3/CCR4 ratio was significantly lower in the AA patients with extrinsic AD than in those with intrinsic AD (Figure 1f), further confirming the relative Th1 and Th2 dominancy of intrinsic and extrinsic AD, respectively. These results suggest that, in AA patients, Tc1 cells are increased, but the concomitant presence of extrinsic AD skews the balance to type 2 state.

### 2.2. Significant Correlation between SALT and EASI Score in the Patients with AA and Extrinsic AD

The correlation between SALT and EASI score were analyzed in the twelve AA patients with extrinsic AD. Interestingly, the two scores were found to be significantly correlated by Spearman’s correlation coefficient (Figure 2). This result indicates that the improvement of AD may contribute to the severity of AA in the patients with AA and extrinsic AD.

### 2.3. Dual Efficacy of Dupilumab for AD and AA in Extrinsic AD Patients

Six AA patients with extrinsic AD were treated with dupilumab (Table 1). As shown in a representative case, dupilumab exerted a therapeutic effect on not only AD (EASI score, from 30.0 to 0.50; Figure 3a,b) but also AA (SALT score, from 100 to 42; Figure 3c,d) at 6 months after initiation of the therapy. In the 6 patients, remarkable AD improvement and hair regrowth were found after 6 months of treatment, as the average EASI score of AD (Figure 3e) and SALT score of AA (Figure 3f) were significantly decreased in parallel. However, in a patient with intrinsic AD, dupilumab had no effect on AA (SALT score, from 100 to 100), although the patient’s AD was markedly improved (EASI score, from 35.0 to 2.0) by the therapy (Figure 3g–j). The alterations of circulating T cells were assessed by CXCR3 and CCR4 expressed on Th1/Tc1 and Th2/Tc2 cells, respectively. The CXCR3/CCR4 ratio in the AA patients with extrinsic AD was increased after dupilumab treatment compared to that of the pre-treatment (Figure 3k).

### 2.4. Improvement of Histopathological Changes by Dupilumab in AA with Extrinsic AD

The histopathology of AA is characterized by aggregation of lymphocytes in and around hair bulbs in skin lesions, which is so-called “swarm of bees” appearance [31]. There have been very few reported cases showing eosinophil infiltration in both transverse and vertical sections, presumably because AA is a virtually type 1-preponderant inflammatory disease [32]. In an AA case with extrinsic AD (case 2), we conducted a histological study. In HE staining, there was a dense infiltrate of not only lymphocytes, but also eosinophils (Figure 4a). Dupilumab treatment decreased the numbers of lymphocytes and eosinophils infiltrating around hair bulbs (Figure 4b). An immunostaining of the pre-treatment skin specimen revealed that CCR4^+^ cells (Figure 4c) as well as CXCR3^+^ cells (Figure 4d) infiltrated around hair bulbs. After 16 weeks of dupilumab treatment, less infiltration of CCR4^+^ cells (Figure 4e), but retained infiltration CXCR3^+^ cells, was found (Figure 4f). Numerical assessment showed that the number of CCR4^+^ cells was significantly decreased after the therapy (0.22 ± 0.15) (*p* = 0.021) compared to that before the therapy (1.11 ± 0.39) (Figure 4g), whereas CXCR3^+^ cells was not changed (Figure 4h). These results suggest that dupilumab therapy shifts the immunological condition of the skin lesion from the type 2 to the type 1 state.

## 3. Discussion

It has been thought that AA is a type 1 inflammatory disease with a crucial role of IFN-γ [10]. Although AA is complicated with AD at a rather high frequency, AD is primarily a Th2-driven disease with increased levels of Th2 cytokines [17]. Therefore, trichologists have shied away from confronting and understanding this contradiction. Recently recognized classification of AD into the extrinsic and intrinsic types urges us to further understand the AD pathogenesis and to select the treatments in individual AD patients [21]. The immunological condition of AA may be different between atopic and non-atopic patients and between extrinsic and intrinsic AD patients. In this context, it is an interesting observation to evaluate the therapeutic effect of dupilumab on AA. The patterns of AA responsiveness to dupilumab are various in the reported cases, as AA was improved in 7 cases [11,16,25,26,27,28,29], AA was developed in 8 cases [13,14,15,27,33,34,35,36], and AA progressed in 2 cases [34,36]. In the present study, we had 7 AA patients with AD who were treated with dupilumab at our hospital: 6 extrinsic AD cases and one intrinsic AD case. While dupilumab improved both AD and AA in all extrinsic AD patients, it alleviated AD, but not AA, in the intrinsic AD patient.

Our study on the circulating T cell populations, the Th1 to Th2 ratio, as assessed by IFN-γ^+^CD4^+^/IL-4^+^CD4^+^ ratio and IFN-γ^+^CD4^+^/IL-13^+^CD4^+^ ratio, was significantly higher in non-atopic AA than in extrinsic AD. This is an expected finding, because type 1 and type 2 preponderance is well known in AA and extrinsic AD, respectively. When evaluated with the alternative markers, the Th1/Tc1 (CXCR3^+^) to Th2/Tc2 (CCR4^+^) ratio was significantly higher in AA with intrinsic AD than in AA with extrinsic AD. This high Th1/Tc1 frequency may reflect the relative Th1 skewing condition of intrinsic AD [21] in addition to the type 1 shift of AA. The notion that the co-existence of extrinsic AD induces a relative type 2 skewing was supported by the histological finding that CCR4^+^ cells infiltrated around hair bulbs in AA lesions. While the frequency of IFN-γ^+^ cells inversely correlated with the IgE values in non-atopic AA patients, such correlation was not found in AA patients with extrinsic AD, suggesting vague influence of AD status in AA patients complicated with AD. These results are supported by the significant correlation between SALT and EASI score indicated in Figure 2. In the AA patients with extrinsic AD, along with the clinical improvement by dupilumab therapy, circulating CXCR3^+^ cells remained unchanged and CCR4^+^ cells were decreased in number. Given that Th2/Tc2 as well as Th1/Tc1 play a role for the development of AA, dupilumab might improve both AD and AA by depressing the type 2 inflammation. While non-atopic AA shows merely type 1 inflammation [37], Th2/Tc2 cells are involved in AA with extrinsic AD.

The efficacy of dupilumab for AA with AD has been discussed in recent review papers [38,39]. AD patients hold a 26-fold greater risk of concomitant with AD compared to health control [40]. Therefore, these two diseases share Th2/Tc2-shifted immune conditions with upregulation of IL-4 and IL-13. On the other hand, there are also several case reports of the induction of AA by the treatment with dupilumab for AD [41]. This can be explained that dupilumab may improve AA with AD but induces AA in the patients with AD without AA [34]. McKenzie et al. has used dupilumab for 16 pediatric AA patients with concomitant A [42]. Most had longstanding disease (median time from diagnosis 4 years) and were refractory to multiple therapies prior to dupilumab. In the study, dupilumab showed greater likelihood of regrowth in the patient with more severe and longstanding histories of AD.

Contact immunotherapy is highly recommended as a treatment of AA. Although the mechanism underlying its effectiveness is still not fully understood, its modification of immune balance may contribute to the therapeutic improvement [43]. Repeated elicitation of contact hypersensitivity by a hapten induces a shift in the cutaneous cytokine milieu from a Th1 to a Th2 profile [44], thereby downregulating the type1 reaction and exerting a possible therapeutic efficacy in AA lesions. However, contact immunotherapy is not effective for AA with ordinary AD [45]. Probably, this immunotherapy may exaggerate AD-associated type 2 polarization in the patients. Given that not only type 1 but also type 2 reactions are involved in AA of the patients with extrinsic AD, anti-IL-4/IL-13 mAb has a potential to improve AA. A couple of studies have shown that AA with a poor response to topical immunotherapy showed increased IL-4 production after the treatment [46,47]. Although the patient number is limited, it may be reasonable that our case of AA with intrinsic AD did not respond to dupilumab. Our study provides an important notion that type 2 immunity may participate in the development of AA in AD patients. It is considered that the immunological state of non-atopic AA is different from that of atopic AA.

## 4. Materials and Methods

### 4.1. Patients and Samples

Peripheral blood mononuclear cells (PBMCs) were taken from 11 non-atopic patients with chronic-phase AA (hair loss for longer than 6 months and Severity of Alopecia Tool (SALT) score > 25), 13 patients with chronic-phase AA and extrinsic AD, 6 patients with chronic-phase AA and intrinsic AD, and 9 healthy volunteers.

PBMCs were isolated from heparinized venous blood by centrifugation with Ficoll-Paque PLUS in LeucoSep tubes (Greiner Bio-One, Frickenhausen, Germany). Skin samples were obtained from one AA patient with extrinsic AD.

### 4.2. Dupilumab Treatment

Six AA patients with extrinsic AD and one AA patient with intrinsic AD were treated with dupilumab. Dupilumab was initiated at 600 mg subcutaneously, followed by 300 mg every 2 weeks. The patients were treated for 6 months to 1 year.

### 4.3. Eczema Area and Severity Index (EASI) Score and Severity of Alopecia Tool (SALT) Score

EASI score was measured as reported previously [48,49]. The degree of hair loss based on the percentage of the affected scalp surface area was assessed by SALT score [50]. The scalp was divided into four parts according to the surface area as follows: the vertex or top = 40% (0.40), right side = 18% (0.18), left side = 18% (0.18), and the posterior aspect = 24% (0.24). The percentage of hair loss in any of the four areas was multiplied by the percentage of the scalp covered in that area. The SALT score is the sum of the percentages of hair loss in all of the areas [SALT score = (percentage of hair loss at the vertex × 0.4) + (percentage of hair loss at the right side × 0.18) + (percentage of hair loss at the left side × 0.18) + (percentage of hair loss at the posterior aspect × 0.24)].

### 4.4. Flow Cytometric Analysis

The cell surface expression of CD4, CD8, CXCR3, and CCR4 on PBMCs was determined by flow cytometric analysis. PBMCs were collected from 7 acute-phase AA patients, 7 chronic-phase AA patients, and 7 healthy subjects. Briefly, PBMCs were incubated with PerCP-conjugated mouse anti-human CD4 and CD8 monoclonal antibodies (mAbs; BD PharMingen, San Diego, CA, USA), FITC-conjugated mouse anti-human CXCR3 mAb (BD PharMingen), and PE-conjugated mouse anti-human CCR4 mAb (BD PharMingen). After washing, the cells were subjected to flow cytometric analysis (FACSCanto II, Becton Dickinson, Franklin Lakes, NJ, USA). Dead cells were excluded by adding 7-amino-actinomycin D (7-AAD).

### 4.5. Intracytoplasmic Cytokines

The intracytoplasmic expression of IFN-γ, IL-4, IL-13, and IL-17 was analyzed by flow cytometric analysis. PBMCs (2 × 10^6^ cell/mL) were stimulated with 25 ng/mL phorbol 12-myristate 13-acetate (PMA; Sigma, St. Louis, MO, USA) and 1 μg/mL ionomycin for 4 h. Then, 1 μL of GolgiPlug (BD PharMingen) was added for every 1 mL of cell culture. The stimulated PBMCs were subsequently stained with PerCP-conjugated mouse anti-human CD4 mAb or anti-human CD8 mAb (BD PharMingen) for 30 min. After washing with FACS buffer, the cells were thoroughly resuspended and 250 μL of Fixation/Permeabilization solution (BD Biosciences, San Diego, CA, USA) was added to the tubes (BD Biosciences). The cells were incubated for 20 min at 4 °C and washed twice. Then, intracellular cytokines were stained with FITC-conjugated mouse anti-human IFN-γ mAb (BD Biosciences), PE-conjugated anti-human IL-4 mAb (BD Biosciences), or PE-conjugated anti-human IL-13 mAb (BD Bioscience) for 30 min. After washing, stained cells were used for flow cytometric analysis on FACSCanto II.

### 4.6. Histopathology and Immunohistochemical Staining

Formalin-fixed and paraffin-embedded human scalp skin samples were deparaffinized and stained by hematoxylin and eosin (H&E). They were also immunostained with mouse anti-human CXCR3 or anti-human CCR4 mAb (Abcam, Cambridge, UK), or mouse anti-human CD4 or anti-CD8 mAb (DAKO, Glostrup, Denmark), as reported previously [37]. The reactivity was visualized by AEC staining (Nichirei Bioscience, Tokyo, Japan). Regarding CXCR3-positive cells and CCR4-positive cells infiltrating around the hair follicles, the number of cells at 10 locations in the frame of 200 μm square around the hair follicles was examined.

### 4.7. Statistical Analysis

Each value is expressed as the mean ± standard deviation (SD) or the mean ± standard error (SE). Data were analyzed by the Kruskal–Wallis test and paired/unpaired student’s *t* test using GraphPad Prism (GraphPad Software, San Diego, CA, USA). Differences were considered to be statistically significant at *p* < 0.05. Pearson correlation analysis was used to investigate the correlation between EASI score and SALT score.

## Figures and Tables

**Figure 1 ijms-22-02618-f001:**
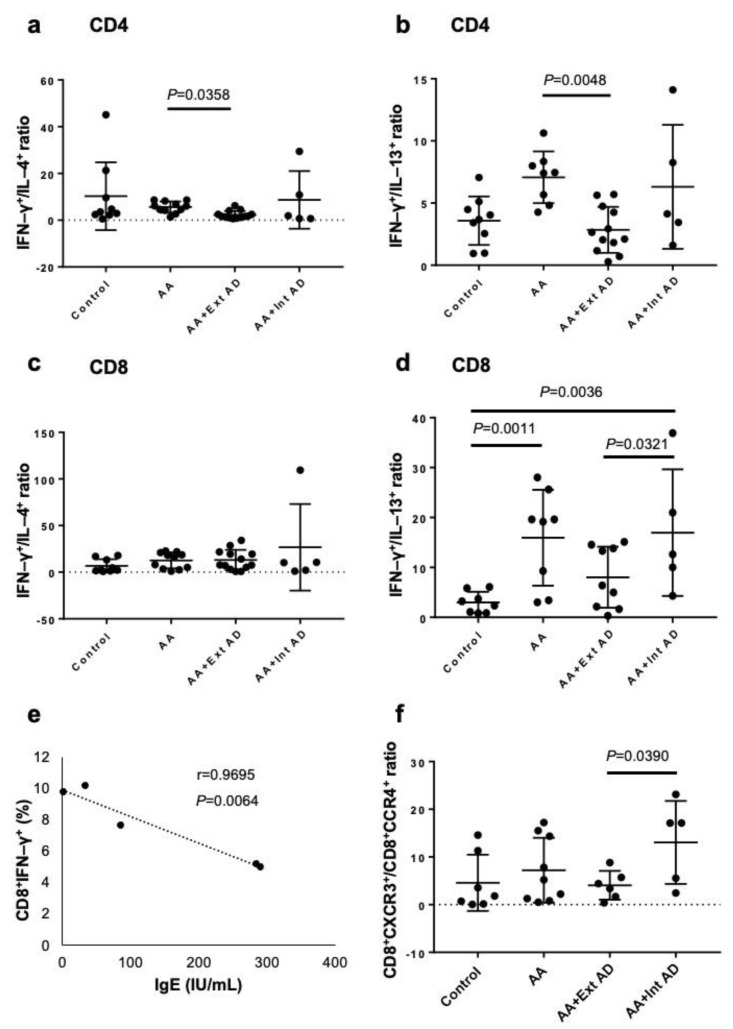
Ratio of Th1/Tc1 to Th2/Tc2 in peripheral blood of non-atopic and atopic patients with AA. (**a**–**d**) The expression of cytoplasmic cytokines was analyzed by flow cytometric analysis. The ratios of IFN–γ/IL-4 and IFN–γ/IL-13 in CD4^+^ (**a**,**b**) and CD8^+^ T cells (**c**,**d**) were measured in normal control subjects, non-atopic AA, and AA with extrinsic or intrinsic AA. (**e**) The correlation between the frequency of IFN–γ-producing cells and the serum IgE level was analyzed in the patients with non-atopic AA. (**f**) The ratio of CD8^+^CXCR3^+^ (Tc1) to CD8^+^CCR4^+^ (Tc2) cells was analyzed by flow cytometric analysis in normal control subjects, non-atopic AA, and AA with extrinsic or intrinsic AA.

**Figure 2 ijms-22-02618-f002:**
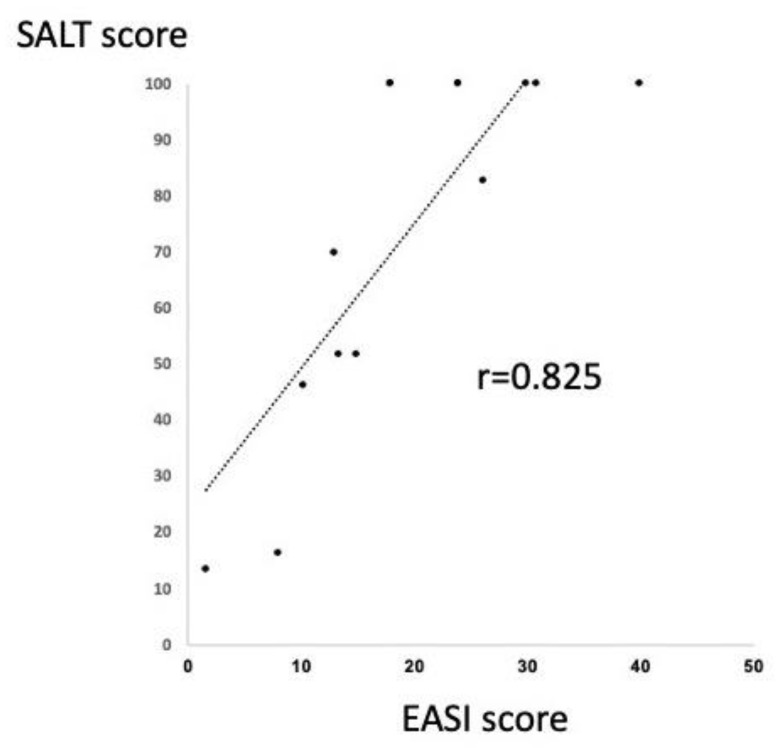
Significant correlation between SALT and EASI score in the patients with AA and extrinsic AD. The correlation between SALT and EASI score were analyzed in the twelve AA patients with extrinsic AD. Interestingly, the two scores were found to be significantly correlated by Spearman’s correlation coefficient.

**Figure 3 ijms-22-02618-f003:**
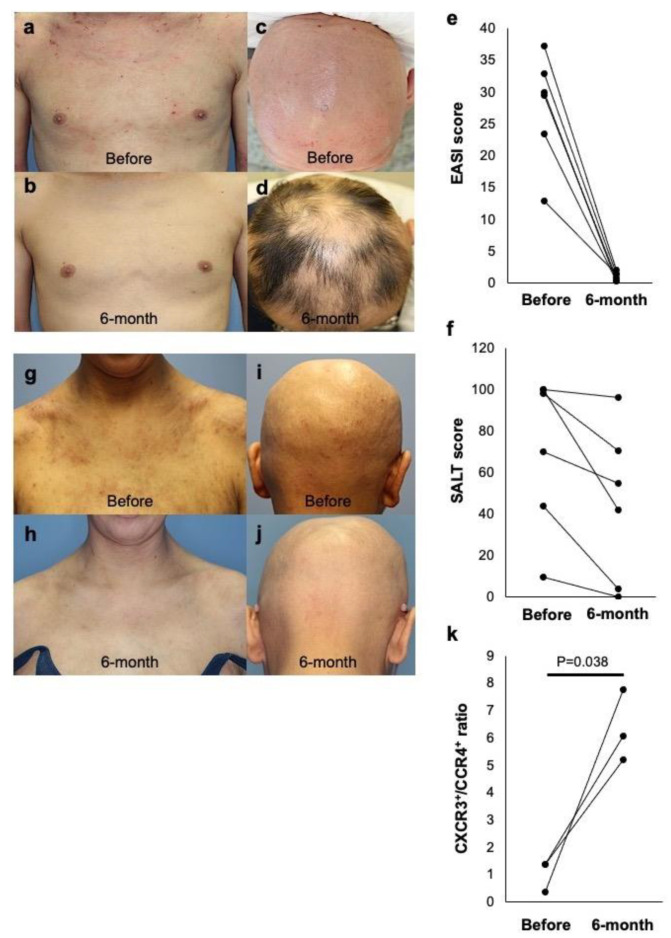
Clinical improvement of AD and AA by dupilumab treatment. (**a**–**d**) Representative case of AA with extrinsic AD before and 6 months after dupilumab therapy. (**e**,**f**) Changes of EASI scores (**e**) and SALT scores (**f**) in 6 patients with extrinsic AD and AA before and 6 months after dupilumab therapy. (**g**–**j**) Representative case of AA with extrinsic AD before and 6 months after dupilumab therapy. (**k**) The CXCR3/CCR4 ratio in the AA patients with extrinsic AD.

**Figure 4 ijms-22-02618-f004:**
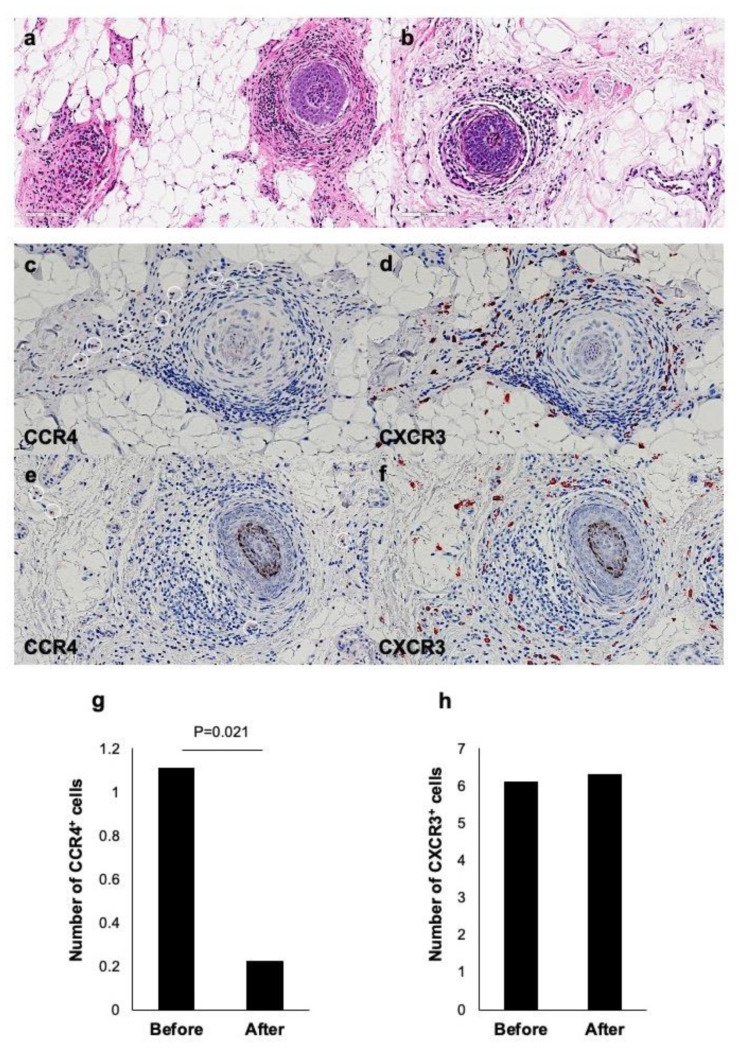
Histological and immunohistochemical findings in a case of AA with extrinsic AD (case 2). H.E. staining before (**a**) and 16 weeks after (**b**) dupilumab therapy. Immunostaining of CCR4 and CXCR3 before (**c**,**d**) and 16 weeks after (**e**,**f**) dupilumab therapy. (**g**,**h**) The number of CCR4 negative and CXCR3 positive cells before and 16 weeks after dupilumab therapy.

**Table 1 ijms-22-02618-t001:** Summary of 7 cases of AA with AD treated with dupilumab.

Cases	Age	Sex	Tx Period (Week)	Type of AD	IgE Level before Tx (IU/mL)	IgE Level at 20 Weeks (IU/mL)	EASI Score before Tx	EASI Score after Tx	SALT Score before Tx	SALT Score after Tx	Clinical Outcome
1	46	M	70	Extrinsic	58737	8577	32.9	1.5	9.4	0	Hair regrowth at 12 weeks.
2	35	M	70	Extrinsic	5869	1207	29.9	0.4	100	42.0	Hair regrowth at 8 weeks.
3	47	M	66	Extrinsic	10522	4634	37.3	2.1	100	96.0	Slight hair growth at 26 weeks.
4	48	F	64	Extrinsic	14665	6496	29.4	0.8	98.0	70.4	Hair regrowth at 24 weeks.
5	48	F	58	Extrinsic	3152	1442	23.4	0.4	44.0	4.0	Hair regrowth at 12 weeks.
6	30	M	40	Extrinsic	7372	4800	13.0	1.5	70.0	54.8	Hair regrowth at 16 weeks.
7	38	F	70	Intrinsic	122	22	35.0	1.0	100	100	No hair regrowth.

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
