# Peer review of "Immunological Properties of Atopic Dermatitis-Associated Alopecia Areata"

_ijms, 2021, doi:10.3390/ijms22052618_

Round 1
Reviewer 1 Report
The manuscript : "Immunological properties of atopic dermatitis-associated alopecia areata" is interesting analyze of coincidence AA with atopic dermatitis. The authors described promissing effect of dupilumab treatment.
The discussion based on the references 1096-2019. The authors should improve the discussion and considered literature from last year:
Pourang A, Mesinkovska NA. New and Emerging Therapies for Alopecia Areata. Drugs. 2020 May;80(7):635-646. doi: 10.1007/s40265-020-01293-0. PMID: 32323220.
Żeberkiewicz M, Rudnicka L, Malejczyk J. Immunology of alopecia areata. Cent Eur J Immunol. 2020;45(3):325-333. doi: 10.5114/ceji.2020.101264. Epub 2020 Nov 1. PMID: 33437185; PMCID: PMC7789996.
Lee HH, Gwillim E, Patel KR, Hua T, Rastogi S, Ibler E, Silverberg JI. Epidemiology of alopecia areata, ophiasis, totalis, and universalis: A systematic review and meta-analysis. J Am Acad Dermatol. 2020 Mar;82(3):675-682. doi: 10.1016/j.jaad.2019.08.032. Epub 2019 Aug 19. PMID: 31437543.
Hendricks AJ, Yosipovitch G, Shi VY. Dupilumab use in dermatologic conditions beyond atopic dermatitis - a systematic review. J Dermatolog Treat. 2021 Feb;32(1):19-28. doi: 10.1080/09546634.2019.1689227. Epub 2019 Nov 12. PMID: 31693426.Hendricks AJ, Yosipovitch G, Shi VY. Dupilumab use in dermatologic conditions beyond atopic dermatitis - a systematic review. J Dermatolog Treat. 2021 Feb;32(1):19-28. doi: 10.1080/09546634.2019.1689227. Epub 2019 Nov 12. PMID: 31693426.Hendricks AJ, Yosipovitch G, Shi VY. Dupilumab use in dermatologic conditions beyond atopic dermatitis - a systematic review. J Dermatolog Treat. 2021 Feb;32(1):19-28. doi: 10.1080/09546634.2019.1689227. Epub 2019 Nov 12. PMID: 31693426.Hendricks AJ, Yosipovitch G, Shi VY. Dupilumab use in dermatologic conditions beyond atopic dermatitis - a systematic review. J Dermatolog Treat. 2021 Feb;32(1):19-28. doi: 10.1080/09546634.2019.1689227. Epub 2019 Nov 12. PMID: 31693426.Hendricks AJ, Yosipovitch G, Shi VY. Dupilumab use in dermatologic conditions beyond atopic dermatitis - a systematic review. J Dermatolog Treat. 2021 Feb;32(1):19-28. doi: 10.1080/09546634.2019.1689227. Epub 2019 Nov 12. PMID: 31693426.Hendricks AJ, Yosipovitch G, Shi VY. Dupilumab use in dermatologic conditions beyond atopic dermatitis - a systematic review. J Dermatolog Treat. 2021 Feb;32(1):19-28. doi: 10.1080/09546634.2019.1689227. Epub 2019 Nov 12. PMID: 31693426.
McKenzie PL, Castelo-Soccio L. Dupilumab Therapy for Alopecia Areata in Pediatric Patients with Concomitant Atopic Dermatitis. J Am Acad Dermatol. 2021 Jan 20:S0190-9622(21)00195-X. doi: 10.1016/j.jaad.2021.01.046. Epub ahead of print. PMID: 33484763.
Author Response
Answer to Reviewer 1
The discussion based on the references 1096-2019. The authors should improve the discussion and considered literature from last year:
⇒Thank you for your kind suggestion with the following new paper on the new treatments for AD. The following papers have been cited, and the discussion has been revised based on the added papers.
Pourang A, Mesinkovska NA. New and Emerging Therapies for Alopecia Areata. Drugs. 2020 May;80(7):635-646. doi: 10.1007/s40265-020-01293-0. PMID: 32323220.
Żeberkiewicz M, Rudnicka L, Malejczyk J. Immunology of alopecia areata. Cent Eur J Immunol. 2020;45(3):325-333. doi: 10.5114/ceji.2020.101264. Epub 2020 Nov 1. PMID: 33437185; PMCID: PMC7789996.
Lee HH, Gwillim E, Patel KR, Hua T, Rastogi S, Ibler E, Silverberg JI. Epidemiology of alopecia areata, ophiasis, totalis, and universalis: A systematic review and meta-analysis. J Am Acad Dermatol. 2020 Mar;82(3):675-682. doi: 10.1016/j.jaad.2019.08.032. Epub 2019 Aug 19. PMID: 31437543.
Hendricks AJ, Yosipovitch G, Shi VY. Dupilumab use in dermatologic conditions beyond atopic dermatitis - a systematic review. J Dermatolog Treat. 2021 Feb;32(1):19-28. doi: 10.1080/09546634.2019.1689227. Epub 2019 Nov 12. PMID: 31693426.
McKenzie PL, Castelo-Soccio L. Dupilumab Therapy for Alopecia Areata in Pediatric Patients with Concomitant Atopic Dermatitis. J Am Acad Dermatol. 2021 Jan 20:S0190-9622(21)00195-X. doi: 10.1016/j.jaad.2021.01.046. Epub ahead of print. PMID: 33484763.

Reviewer 2 Report
Very interesting paper highlighting that type 2 immunity may participate in the development of Alopecia Areata in extrinsic Atopic Dermatitis patients;Article will be eligible to publication after a round of review.
I have some queries:
in the materials and methods 4.7 statistical analysis section: Please specify what kind of statistical program was used in order to perform the analysis and its maker and city.
In the introduction section, you only nominate dupilumab not specifying its mechanism of action; I would better clarify the innovative treatments available for AD; here an interesting reference doi: 10.1111/dth.12787.
page 2 line 61-63 this sentence needs a reference; I think that this article would be a perfect fit doi: 10.18176/jiaci.0519.
Thank You
Author Response
Answer to Reviewer 2
Very interesting paper highlighting that type 2 immunity may participate in the development of Alopecia Areata in extrinsic Atopic Dermatitis patients; Article will be eligible to publication after a round of review.
⇒Thank you for your interest in our study on immunological properties in patients with AA and AD.
In the materials and methods 4.7 statistical analysis section: Please specify what kind of statistical program was used in order to perform the analysis and its maker and city.
⇒Thank you for your important notice about statistical program that is missing in present manuscript. We used GraphPad Prism for statistical analysis in this study. The company of GraphPad Prism is GraphPad Software, San Diego, CA, USA. The name of maker and its city have been added in the section of materials and methods followed by reviewer’s kind suggestion.
In the introduction section, you only nominate dupilumab not specifying its mechanism of action; I would better clarify the innovative treatments available for AD; here an interesting reference doi: 10.1111/dth.12787.
⇒Thank you for your kind suggestion about this excellent paper, Dattola et al. What’s new in the treatment of atopic dermatitis? Dermatol Ther 2019; 32: e12787. As reviewer indicates, the innovative treatments, such as tofacitinib, baricitinib, crisaborole, nemolizumab, apremilast and Lebrikizumab, have been available for severe AD. As reviewer kindly suggested, these treatments have been introduced in the introduction section in addition to dupilumab.
Page 2 line 61-63 this sentence needs a reference; I think that this article would be a perfect fit doi: 10.18176/jiaci.0519.
⇒Thank you for your excellent suggestion about the following paper to fit page 2 line 61-63: Nettis et al. A multicentric study on prevalence of clinical patterns and clinical phenotypes in adult atopic dermatitis. J Investig Allergol Clin Immunol 2020; Vol. 30(6). The paper has been added in the revised paper.

Round 2
Reviewer 2 Report
The authors responded to all queries. The article is ready to be published